# MAGIs regulate aPKC to enable balanced distribution of intercellular tension for epithelial sheet homeostasis

Kenji Matsuzawa [1], Hayato Ohga[1], Kenta Shigetomi[1], Tomohiro Shiiya[2], Masanori Hirashima[2] & Junichi Ikenouchi [1✉]

Constriction of the apical plasma membrane is a hallmark of epithelial cells that underlies cell shape changes in tissue morphogenesis and maintenance of tissue integrity in homeostasis. Contractile force is exerted by a cortical actomyosin network that is anchored to the plasma membrane by the apical junctional complexes (AJC). In this study, we present evidence that MAGI proteins, structural components of AJC whose function remained unclear, regulate apical constriction of epithelial cells through the Par polarity proteins. We reveal that MAGIs are required to uniformly distribute Partitioning defective-3 (Par-3) at AJC of cells throughout the epithelial monolayer. MAGIs recruit ankyrin-repeat-, SH3-domain- and proline-rich-region-containing protein 2 (ASPP2) to AJC, which modulates Par-3-aPKC to antagonize ROCK-driven contractility. By coupling the adhesion machinery to the polarity proteins to regulate cellular contractility, we propose that MAGIs play essential and central roles in maintaining steady state intercellular tension throughout the epithelial cell sheet.

[1] Department of Biology, Faculty of Sciences, Kyushu University, 774 Motooka, Nishi-ku, Fukuoka 819-0395, Japan. [2] Division of Pharmacology, Graduate School of Medical and Dental Sciences, Niigata University, 1-757 Asahimachi-dori, Chuo-ku, Niigata 951-8510, Japan. ✉email: ikenouchi.junichi.033@m.kyushu-u.ac.jp

Epithelial tissues expand and contract during development and in homeostasis, enabling them to take on complex topologies to seal the interior of the organism from the external environment[1]. Tissue integrity is maintained by modulating the inherent contractility of each epithelial cell that comprises the tissue. Specifically, a contractile actomyosin network is tethered to the plasma membrane by the apical junctional complexes (AJC), claudin-based tight junctions (TJ) and cadherin-based adherens junctions (AJ), to generate a constitutive tensile force. Since epithelial cells are mechanically coupled by the AJC, individual tensile forces are transmitted and shared over multiple neighboring cells to dissipate tension in the epithelial cell sheet or to potentiate localized tissue deformation at nodes[2–4]. Thus, it is crucial to strictly control contractile activity at apical cell adhesions, whether the end goal is to maintain the integrity of the epithelial cell sheet or to induce its deformation.

Signaling through RhoA regulates apical cell contractility, in part through a feedback mechanism involving non-muscle myosin (NM) II isoforms[5,6]. RhoA and its effector Rho-associated, coiled-coil-containing protein kinase (ROCK) are required to concentrate NMIIA at AJ[7]. It is critical to precisely regulate ROCK at AJC, whether to alter or to maintain the gross morphology of an epithelial tissue. For example, deformation of the epithelial monolayer, such as during neural tube closure, requires constriction of specific cells within the cell sheet by upregulating the circumferential actomyosin network at AJC. In the mouse[8] and chicken[9] embryos, elevation of gene expression level of Shroom is required to recruit ROCK to apical junctions of neuroepithelial cells that are located at the hinge region and induce apical constriction to drive neural tube closure; forced expression of Shroom in cultured epithelial cells increases junctional myosin activation and causes cells with a broad range of apical cell areas to dot the cell sheet[8].

By contrast, in the case of planer epithelial cell sheets, the apical contractile forces of the constituent cells must be balanced and maintained within a certain range so to avoid unnecessary sheet deformation. Recently, it was reported that the Par polarity proteins are involved in this regulation. Par-3, together with the FERM-domain protein Willin, suppresses the junctional localization of ROCK by recruiting aPKC[10]. aPKC phosphorylates ROCK and suppresses its junctional localization, thereby allowing cells to retain uniformly shaped apical domains. Loss of either Par-3 or aPKC leads to apical constriction and abnormal epithelial apical morphology. Therefore, in order to adjust the tension among nearby cells in the epithelial sheet, it is necessary to control the amounts of Par-3 and aPKC at AJC through some cell-adhesion-associated proteins. However, such molecular mechanism has not been described.

In this paper, we clarified that the cell adhesion-related molecules MAGI-1 and MAGI-3 contribute to the steady state level of apical domain contractility by mobilizing a complex of signaling proteins that culminate in aPKC-mediated antagonism of junctional ROCK activity. MAGI proteins are highly conserved members of the membrane-associated guanylate kinase protein superfamily[11]. They were identified as junctional constituents that promote AJ assembly. In mammalian cells, MAGI-1 and MAGI-2 presumably localize to AJ via β-catenin, and MAGI-1 was shown to positively regulate AJ formation in endothelial cells[11–13]. The *C. elegans* MAGI ortholog localizes apically to cadherin-based adhesions and its loss leads to actin disorganization and reduces the overall robustness of cell adhesions in the embryonic epidermis[14,15]. In addition to their role in supporting junctional architecture, MAGIs also interact with signaling molecules such as the phosphatases PTEN and receptor tyrosine phosphatase β, suggesting that they can function as signaling modulators at AJC[16,17]. As such, the precise roles of MAGI at AJC remain to be elucidated.

Here, we propose a molecular mechanism by which AJC scaffolding proteins control apical cell contractility by differentially recruiting MAGI-1 and MAGI-3 to apical junctions. MAGIs further localize an array of scaffolding and signaling proteins that recruit and regulate Par-3 function to modulate contractility of the AJC-linked actomyosin network. Thus, we revealed the MAGIs are essential regulators of Par polarity proteins that are central to the regulation of tension distribution in epithelial tissue homeostasis.

## Results

**Loss of ZO proteins strongly perturbs Par-3 localization and alters apical morphology.** We previously showed that depletion of ZO proteins in the mouse mammary epithelial cell line, EpH4, delays the formation of the contractile belt-like AJ[18], suggesting that ZO proteins are required for epithelial polarization. In the course of our investigation, we noted greater irregularity in shape and size of the apical region in ZO-1 and ZO-2 double knockout (ZO-1,-2 DKO) cells in comparison with parental (WT) EpH4 cells (Fig. 1a). The WT cell sheet was composed of cells that were generally of the same size and the lengths of cell junctions in each cell showed high uniformity. Meanwhile, the ZO-1,-2 DKO monolayer was an admixture of comparatively smaller and larger cells, each with cell junctions that showed high variability in their lengths. These observations suggested to us that apical cell junctions of cells in the ZO-1,-2 DKO cell sheet were subjected to unbalanced tensile strain from surrounding cells. We observed elevated immunofluorescence intensity of the α18 antibody staining, which is specific to activated α-catenin conformation under tensile strain, in ZO-1,-2 DKO cells (Supplementary Fig. S1a, b). Moreover, intercellular gaps were frequently observed at tricellular contact sites in ZO-1,-2 DKO cells, indicating dysregulation and excess of contractile activity throughout apical junctions (Supplementary Fig. S1a, insets). In support of this, we found that perijunctional myosin II activation was prominent in ZO-1,-2 DKO but not in parental (WT) EpH4 cells (Fig. 1b, c).

In order to investigate the effect of ZO depletion on cell shape as well as to examine protein recruitment to AJC in greater detail, we performed cell segmentation based on activated α-catenin staining (Supplementary Fig. S1c). When we compared the mean apical areas extracted from such contours, we found that there was no significant difference between WT and ZO-1,-2 DKO cells (Fig. 1d, e). We saw that the apical area in WT cells showed a normal distribution (median D'Agostino & Pearson normality test $P$ value = 0.2670 where $P > 0.05$ strongly suggests normal distribution); by contrast, ZO-1,-2 DKO cells did not conform to a normal distribution ($P = 0.0322$; Fig. 1f, g). Treatment of ZO-1,-2 DKO cells with the ROCK inhibitor Y-27632 restored the normal distribution of apical cell area ($P = 0.2921$), suggesting that aberrant ROCK activation could be inducing elevated contractile activity at AJC to cause the irregular apical morphology (Fig. 1d–g). We found that ROCK1 was delocalized from the lateral and apical membranes and was instead strongly concentrated at AJC where it colocalized with activated α-catenin in ZO-1,-2 DKO cells (Fig. 1h, i).

We next examined Par-3 since its localization downregulates junctional ROCK[10]. Par-3 fluorescence was uninterrupted throughout the apical junction in WT cells. By contrast, Par-3 presented discontinuously in ZO-1,-2 DKO cells; there were frequent gaps spanning as much as several micrometers at bicellular junctions in addition to the expected gaps at tricellular contacts (Fig. 1j). Assessing the junctional coverage of Par-3 against total AJC area revealed that Par-3 was significantly less

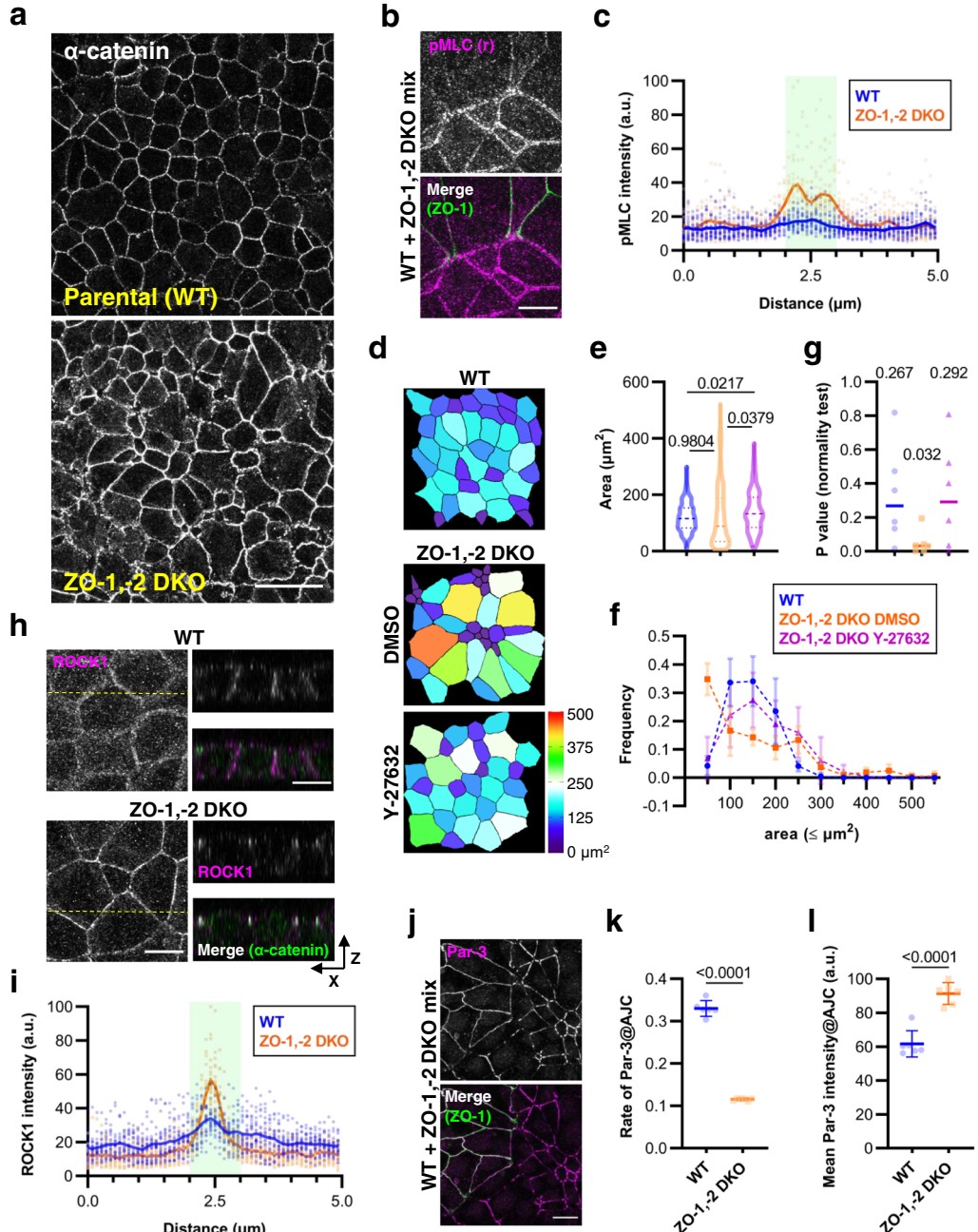

likely to localize to AJC in ZO-1,-2 DKO cells compared with WT cells (Fig. 1k). However, Par-3 was more concentrated when it was localized at cell junctions in ZO-1,-2 DKO cells, as previously reported by Choi et al[19]. (Fig. 1l). Although ZO-1,-2 DKO cells can form E-cadherin-based AJs normally as previously reported[20], the tensional homeostasis at AJC is impaired and cell polarization is abnormal from the viewpoint of subcellular localization of Par-3. Altogether, these results suggest that the TJ scaffolding proteins ZO-1 and ZO-2 regulate apical cell size by modulating ROCK-induced contractile activity through Par-3.

**AJC scaffolding proteins regulate Par-3 localization through MAGIs.** That Par-3 was intermittently targeted to AJC in ZO-1,-2 DKO cells indicated to us that ZO proteins are somehow required for the robustness of junctional Par-3 at the population level in the epithelial monolayer. How do ZO proteins promote Par-3 localization at AJC? It was previously reported that the Par-3 localization at AJC? It was previously reported that the Par-3

targets to TJ by binding to the membrane protein junctional adhesion molecule (JAM)-A[21,22]. However, since junctional localization of JAM-A is completely lost in cells lacking ZO proteins[23], we reasoned that other molecules must be required to uniformly distribute Par-3 at AJC. We looked for adhesion-associated proteins that showed altered localization in ZO-1,-2 DKO cells and identified MAGI proteins as candidate regulators. EpH4 cells express MAGI-1 and MAGI-3 and we found that both proteins localized to AJC in WT cells (Fig. 2a). By contrast, MAGI-3 was absent, and MAGI-1 was diminished, from AJC in ZO-1,-2 DKO cells. Therefore, we considered the possibility that loss of MAGI was responsible for the uneven distribution of Par-3 in these cells. MAGI-1 partially retained its localization in ZO-1,-2 DKO cells, indicating that junctional components other than ZO proteins contribute to its recruitment. Afadin/Canoe is an AJ scaffolding protein that regulates epithelial integrity in parallel with ZO proteins in Drosophila[19,24]. When we examined MAGI

**Fig. 1 Loss of ZO proteins dysregulates ROCK-dependent contractility to alter apical morphology. a** Representative immunofluorescence images of WT and ZO-1,-2 DKO cells stained for activated α-catenin. Scale bar, 20 μm. **b** Representative immunofluorescence images of a co-culture of WT and ZO-1,-2 DKO cells stained for phosphorylated MLC (pMLC, magenta) and ZO-1 (green). Scale bar, 10 μm. **c** Cross-junctional line scans (pMLC) from immunofluorescence images of WT and ZO-1,-2 DKO cells stained for pMLC and activated α-catenin. Shaded area represents AJC as defined by the activated α-catenin peak. Individual data from 20 independent line scans are shown with the means depicted by solid lines. **d** Pseudocolor representations of apical areas in WT and ZO-1,-2 DKO cells. ZO-1,-2 DKO cells were treated with either DMSO or 10 μM Y-27632 for 5 h. Cells were stained for activated α-catenin and processed for area measurement as detailed in Supplementary Fig. S1 and Methods. **e** Violin plot depiction of apical areas in WT and ZO-1,-2 DKO cells. Areas were collated from six measurements obtained over three independent experiments. Lines represent the median and the upper and lower quartiles. $P$ values from Tukey's post-hoc test with one-way ANOVA are shown. $n = 228$ (WT), 219 (ZO-1,-2 DKO DMSO) and 191 (ZO-1,-2 DKO Y27632). **f** Frequency distributions of apical areas in WT and ZO-1,-2 DKO cells. Data were as in **e**. Points are means and error bars are SDs. **g** $P$ values of D'Agostino & Pearson test for normal distribution were computed from the data presented in **f**. Medians are notated and shown graphically as lines. **h** Representative immunofluorescence images of WT and ZO-1,-2 DKO cells stained for ROCK1 (magenta) and activated α-catenin (green). Orthogonal views of the cross-section indicated by the yellow dotted line are shown to the right. Scale bar, 10 μm. **i** Cross-junctional line scans of ROCK1 immunofluorescence from images corresponding to **j**. Shaded area represents AJC as defined by the activated α-catenin peak. Individual data from 20 independent line scans are shown with the means depicted by solid lines. **j** Representative immunofluorescence images of a co-culture of WT and ZO-1,-2 DKO cells stained for Par-3 (magenta) and ZO-1 (green). Scale bar, 10 μm. **k** Quantification of Par-3 junctional coverage relative to total AJC area in WT and ZO-1,-2 DKO cells. Cells were co-stained for Par-3 and activated α-catenin, which was used to designate AJC. $P$ value from unpaired $t$ test is shown. **l** Quantification of Par-3 mean fluorescence intensities at AJC based on the same data as **j**. $P$ value from unpaired t test is shown. Source data are available in Supplementary Data 1.

localization in EpH4 cells in which afadin was knocked out together with ZO proteins (ZO,-1,-2, afadin triple knockout (TKO) cells; Supplementary Figs. S2a, S7), MAGI-1, like MAGI-3 in ZO-1,-2 DKO cells, was lost from AJC (Supplementary Fig. S2b). Crucially, Par-3 no longer accumulated at cell-cell junctions in ZO,-1,-2, afadin TKO cells (Supplementary Fig. S2c). One possible interpretation here is that afadin recruits Par-3 to AJC. However, examination of Par-3 localization in afadin KO cells showed that Par-3 enrichment to AJC was unaltered in the absence of afadin (Supplementary Fig. S2d), suggesting that junctional Par-3 in ZO-1,-2 DKO cells were recruited by remnant MAGI-1.

The stark difference in MAGI-1 and MAGI-3 targeting indicated that these proteins diverge in their affinities for AJC components. We therefore tested the interaction between MAGIs and afadin or ZO proteins by an immunoprecipitation assay in a heterologous expression system. Whereas both MAGI proteins showed similar binding affinities for afadin and ZO-2, MAGI-3 bound ZO-1 with greater efficiency compared to MAGI-1 (Supplementary Figs. S2e–g, S7). We also confirmed that these proteins form physiological complexes in functional epithelial cells (Supplementary Figs. S2h, i, S7). Taken together, these results establish that ZO proteins and afadin cooperatively recruit MAGI proteins to AJC.

We then generated MAGI-1- and MAGI-3-depleted EpH4 cells in order to assess their role in relation to apical morphology (MAGI-1,-3 DKO; Supplementary Figs. S3a, S8). MAGI-1,-3 DKO cells showed no substantial changes in TJ formation as indicated by similarly focused apical Claudin-3 immunofluorescence as in WT cells (Supplementary Fig. S3b). Likewise, apical-basal polarization was unaltered based on similar apical Ezrin and basal-lateral E-cadherin localizations in WT and MAGI-1,-3 DKO cells (Supplementary Fig. S3c). When we looked at apical cell area, we observed a large disparity with no significant difference in mean cell area, as in ZO-1,-2 DKO cells (Fig. 2b, c). ROCK inhibition also mitigated the effect on apical cell area variability, such that MAGI-1,-3 DKO cells showed a normal distribution under this condition ($P_{DMSO} = 0.0499$, $P_{Y-27632} = 0.1212$; Fig. 2b–e). The parallel between these observations and those in ZO-1,-2 DKO cells suggest that MAGI mediate the antagonism of ROCK that was lost in ZO-1,-2 DKO cells. Indeed, we found that ROCK1 localization at AJC was enhanced in MAGI-1,-3 DKO cells, although a substantial amount remained at the lateral membrane (Fig. 2f, g). Perijunctional myosin II

activity was also more pronounced in MAGI-1,-3 DKO cells, reflecting the increased ROCK activity at AJC (Supplementary Fig. S3d, e).

Next, we examined Par-3 localization. Unlike in ZO-1,-2 DKO cells, we did not observe significant regions where Par-3 was altogether absent from AJC in MAGI-1,-3 DKO cells. Instead, we found that Par-3 immunofluorescence was diffuse compared to WT cells and focused apical enrichment could not be observed in examining orthogonal projections (Fig. 2h, i). In line with these observations, both mean Par-3 intensity at AJC and the rate of Par-3 localization at AJC were significantly diminished in MAGI-1,-3 DKO cells, indicative of impaired Par-3 recruitment to AJC (Fig. 2j, k). Thus, MAGIs are required for the proper localization of Par-3 and downregulation of apical cell contractility. It is possible that ZO proteins, afadin, and MAGIs act in parallel to modulate Par-3-dependent ROCK activity. However, our immunofluorescence and interaction studies strongly suggest that MAGIs act downstream of the AJC scaffolds.

**MAGIs control the localization of the Par-3 regulator ASPP2.** Numerous mechanisms are still known to control Par-3 localization aside from MAGIs and JAM-A noted above. Par-3/Baz can be sequestered from cell junctions by phosphorylation-dependent interaction with 14-3-3[25,26]. Phosphorylation can also regulate Par-3 oligomerization, which is essential for proper localization[27,28]. Separately, direct interaction with specific phosphoinositide species is an additional factor in its junctional targeting[29,30]. Some of these mechanisms could account for the divergent effect on Par-3 localization between ZO- and MAGI-depleted cells.

How do MAGI proteins control the targeting of Par-3 to AJC? One intriguing candidate is ASPP2, which was shown to directly bind Par-3 and the loss of which disrupts Par-3 localization at apical junctions in MDCK cells[31]. However, it is unknown how ASPP2 is recruited to AJC. ASPP2 colocalized with MAGI-3 at AJC in WT cells. Intriguingly, when we examined the localization of ASPP2 in MAGI-1,-3 DKO cells, we found that while it retained a membrane localization, its enrichment at AJC was severely reduced (Fig. 3a, b).

Several lines of evidence suggest that another group of proteins, the N-terminal Ras association domain family (RASSF), may mediate MAGIs interaction with ASPP2. The N-terminal RASSF proteins (RASSF7–10) contain an N-terminal Ras association (RA) domain and several coiled coil (CC) domains (Supplementary Fig. S4a). RASSF family proteins have been linked to many

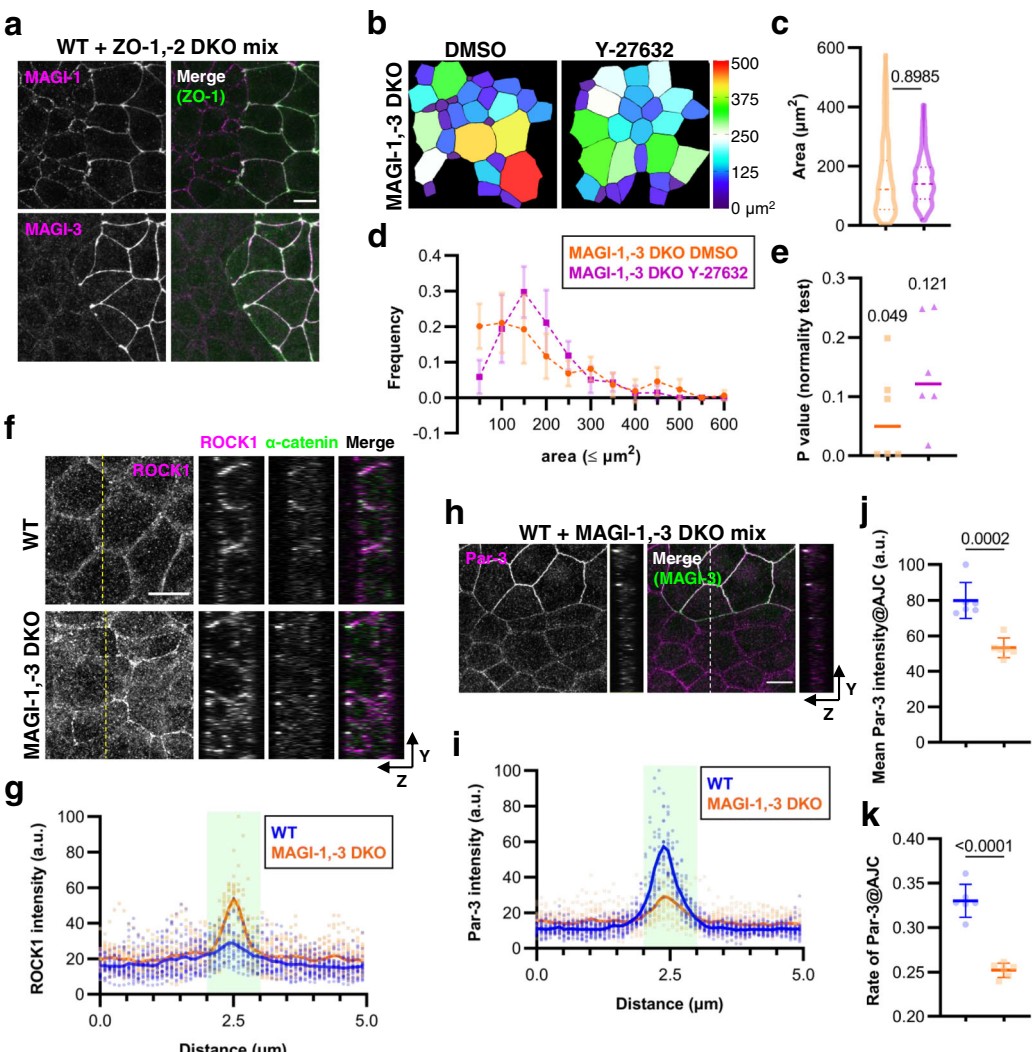

**Fig. 2 MAGI proteins regulate apical contractility and Par-3 localization to AJC. a** Representative immunofluorescence images of a co-culture of WT and ZO-1,-2 DKO cells stained for either MAGI-1 (magenta, upper panel) or MAGI-3 (magenta, lower panel) with ZO-1 (green). Scale bar, 10 μm.
**b** Pseudocolor representations of apical areas in MAGI-1,-3 DKO cells treated with either DMSO or 10 μM Y-27632 for 5 h. **c** Violin plot depiction of apical areas in MAGI-1,-3 DKO cells. Data were collated from six measurements obtained over three independent experiments. Lines represent the median and the upper and lower quartiles. *P* value from unpaired *t* test is shown. *n* = 164 (DMSO) and 147 (Y-27632). **d** Frequency distributions of apical areas in WT and MAGI-1,-3 DKO cells. Data were as in **c**. Points are means and error bars are SDs. **e** P values of D'Agostino & Pearson test for normal distribution were computed from the data presented in **d**. Medians are notated and shown graphically as lines. **f** Representative immunofluorescence images of WT and MAGI-1,-3 DKO cells stained for ROCK1 (magenta) and activated α-catenin (green). Orthogonal views of the cross-section indicated by the yellow dotted line are shown to the right. Scale bar, 10 μm. **g** Cross-junctional line scans of ROCK1 immunofluorescence from images corresponding to **f**. Shaded area represents AJC as defined by the activated α-catenin peak. Individual data from 20 independent line scans are shown with the means depicted by solid lines. **h** Representative immunofluorescence images of a co-culture of WT and MAGI-1,-3 DKO cells stained for Par-3 (magenta) and MAGI-3 (green). Orthogonal views of the cross-section indicated by the white dotted line are shown to the right of each image. Scale bar, 10 μm. **i** Cross-junctional line scans of Par-3 immunofluorescence in WT and MAGI-1,-3 DKO cells. Cells were co-stained with activated α-catenin. Shaded area represents AJC as defined by the activated α-catenin peak. Individual data from 20 independent line scans are shown with the means depicted by solid lines. **j** Quantification of Par-3 mean fluorescence intensities at AJC in WT and MAGI-1,-3 DKO cells from images common with **i**. *P* value from unpaired *t* test is shown.
**k** Quantification of Par-3 coverage relative to total AJC area based on data obtained with **j**. *P* value from unpaired *t* test is shown. Source data are available in Supplementary Data 1.

biological processes and two members are potential tumor suppressors[32]. A proteomic analysis of ASPP binding partners in a human cell line identified RASSF proteins as potential ASPP interactors[33]. In addition, Drosophila RASSF8 and MAGI interact and a loss-of-function RASSF8 mutant that lacks the ability to bind MAGI phenocopies MAGI mutant in Drosophila eye development[34]. Therefore, we designed an immunoprecipitation experiment in a heterologous expression system to examine the possibility that MAGI proteins interact with ASPP via RASSF

proteins. FLAG-tagged ASPP2 was co-expressed with HA-MAGI-3 in addition to either GFP or GFP-RASSF10. HA-MAGI-3 weakly co-precipitated with FLAG-ASPP2 in the presence of GFP alone but more robustly co-precipitated when expressed with GFP-RASSF10 (approximately fourfold over GFP alone), indicating that N-terminal RASSF proteins facilitate MAGI-ASPP interaction (Fig. 3c, d and Supplementary Fig. S7).

Given this finding, we examined RASSF10 interactions with MAGI-3 and ASPP2 more closely. MAGI proteins contain six

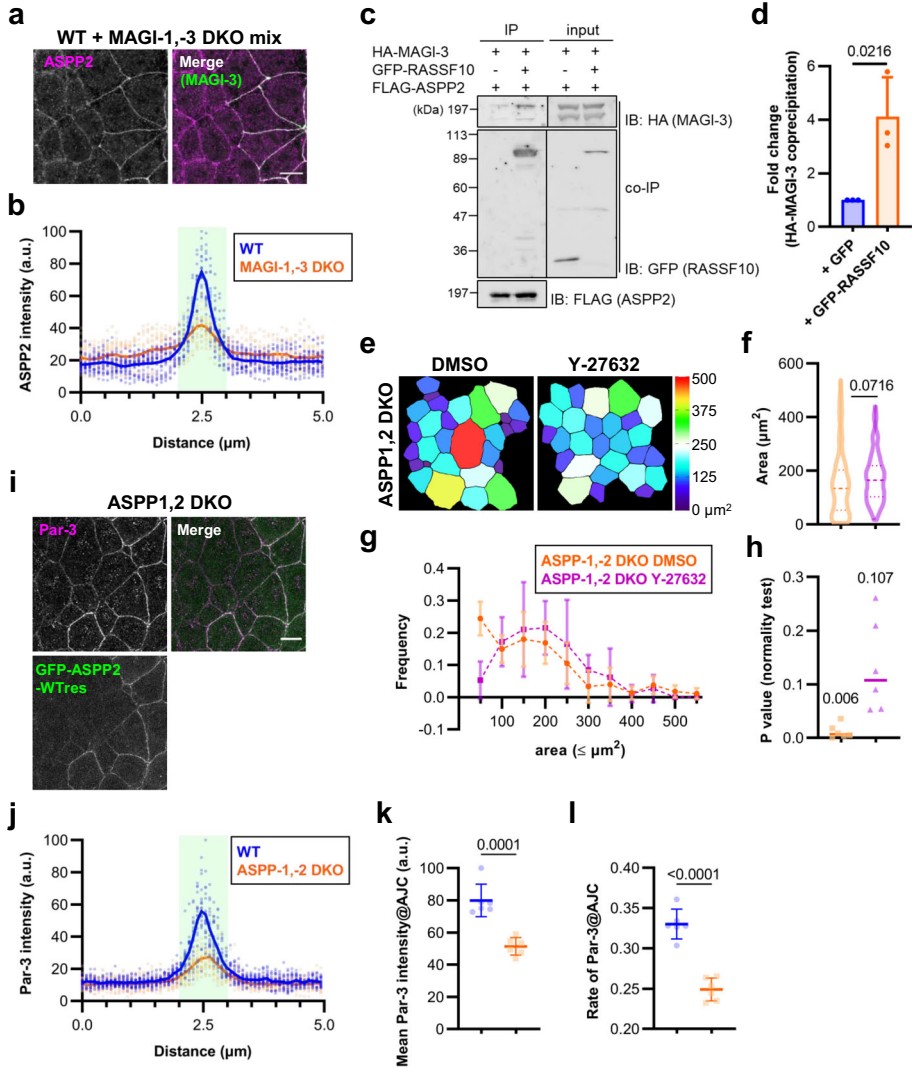

**Fig. 3 ASPP2 is recruited to AJC by MAGIs, where it regulates apical contractility and Par-3 localization to AJC. a** Representative immunofluorescence images of WT and MAGI-1,-3 DKO cells stained for ASPP2 (magenta) and MAGI-3 (green). Scale bar, 10 μm. **b** Cross-junctional line scans from immunofluorescence images of WT and MAGI-1,-3 DKO cells stained for ASPP2 and activated α-catenin. Shaded area represents AJC as defined by the activated α-catenin peak. Individual data from 20 independent line scans are shown with the means depicted by solid lines. **c** Immunoblot of FLAG immunoprecipitation showing interaction between HA-MAGI-3 and FLAG-ASPP2, either with or without GFP-RASSF10. Data is representative of three independent experiments. Uncropped immunoblots are shown in Supplementary Fig. S7. **d** Quantification of **c**. The amount of HA-MAGI-3 coprecipitating with FLAG-ASPP2 in the presence of GFP-RASSF10 was normalized to that of HA-MAGI-3 coprecipitation in the presence of control GFP. *P* value from unpaired *t* test is shown. **e** Pseudocolor representations of apical areas in ASPP1,2 DKO cells treated with either DMSO or 10 μM Y-27632 for 5 h. **f** Violin plot depiction of apical areas in WT and ASPP1,2 DKO cells. Data were collated from six measurements obtained over three independent experiments. Lines represent the median and the upper and lower quartiles. *P* value from unpaired *t* test is shown. *n* = 168 (DMSO) and 131 (Y-27632). **g** Frequency distributions of apical areas in WT and ASPP1,2 DKO cells. Data were as in **f**. Points are means and error bars are SDs. **h** *P* values of D'Agostino & Pearson test for normal distribution were computed from the data presented in **g**. Medians are notated and shown graphically as lines. **i** Representative immunofluorescence images of a co-culture of ASPP1,2 DKO and GFP-ASPP2-WTres cells stained for Par-3 (magenta). Scale bar, 10 μm. **j** Cross junctional line scans of Par-3 immunofluorescence in ASPP1,2 DKO, and GFP-ASPP2-WTres cells. ASPP1,2 DKO cells were co-stained with activated α-catenin. Shaded area represents AJC as defined by either the activated α-catenin or GFP peak. Individual data from 20 independent line scans are shown with the means depicted by solid lines. **k** Quantification of Par-3 mean fluorescence intensities at AJC in WT and ASPP1,2 DKO cells from images common with **j**. *P* value from unpaired *t* test is shown. **l** Quantification of Par-3 coverage relative to total AJC area based on data obtained with **k**. *P* value from unpaired *t* test is shown. Source data are available in Supplementary Data 1.

PDZ domains and both RASSF8 and RASSF10 contain a potential C-terminal PDZ binding motif. We found that full-length RASSF10 (FL) efficiently bound to MAGI-3 but deletion of the C-terminal residues E-S-L-V in RASSF10 (ΔPDZ) abrogated this interaction (Supplementary Figs. S4b, S8). Regarding ASPP2, we determined that the N-terminal half up to CC1 (N half) was

necessary and that CC1 (N2) by itself was sufficient to bind RASSF10 (Supplementary Fig. S4a, c, d, S8).

We wished to probe protein interactions among MAGIs, RASSF and ASPP proteins in intact epithelial cells but several commercial antibodies against RASSF10 failed to detect reliably endogenous level of protein by immunoblotting. Therefore, we

stably expressed FLAG-tagged RASSF10 in EpH4 cells to a low level. MAGI-1, MAGI-3 and ASPP2 coprecipitated with FLAG-RASSF10 (Supplementary Figs. S4e, S8). Taken together, our data indicate that MAGI recruit Par-3 to AJC by controlling ASPP localization, in part through N-terminal RASSF proteins (Supplementary Fig. S4f).

**ASPPs regulate apical contractility and cell shape**. We next generated ASPP1- and ASPP2-depleted EpH4 cells (ASPP1,2 DKO) in order to examine their contribution to cellular contractility and apical cell shape (Supplementary Figs. S5a, S8). The localizations of MAGI-1 and MAGI-3 were unaltered in ASPP1,2 DKO cells, confirming that ASPPs function downstream of MAGIs (Supplementary Fig. S5b). Segmentation analysis showed that, like in ZO-1,-2 DKO, and MAGI-1,-3 DKO cells, apical cell area were highly heterogenous despite little change in mean apical cell area (Fig. 3e, f). Moreover, ROCK inhibition resulted in less varied apical cell area, once again suggesting that ASPPs mediate ZO/afadin-MAGI-dependent ROCK suppression ($P_{DMSO} = 0.006$, $P_{Y-27632} = 0.107$; Fig. 3e–h). Consistent with this data, ROCK1 was highly enriched at AJC and there was a substantial increase in perijunctional/subapical myosin II activation in ASPP1,2 DKO cells in comparison with WT cells (Supplementary Fig. S5c, d). We also found that Par-3 was less concentrated at AJC in ASPP1,2 DKO cells (Fig. 3i–l).

**Antagonism of ROCK activity at AJC requires scaffolding of ASPP2-dependent PP1-aPKC interaction**. The loss of Par-3 from apical junctions led to the up-regulation of ROCK1 activity and constriction of apical domains because the suppression of ROCK1 by aPKC is abolished in the absence of Par-3[10]. We consistently found that Par-3 was displaced from, and ROCK1 was excessively concentrated at, AJC in the KO cells investigated thus far. Our observations indicate that mis-localization of Par-3 allowed for the enrichment of ROCK1 at apical junctions to cause the irregular pattern of apical cell area in the epithelial monolayer. The above findings suggest that the ZO/afadin-MAGI-ASPP signaling axis controls intercellular tension for epithelial sheet homeostasis through junctional targeting of Par-3.

In addition to its role in recruiting Par-3 to TJs, ASPP2 also interacts with protein phosphatase 1 (PP1) and acts as a scaffold to regulate substrate phosphorylation dynamics in many biological contexts such as chromosome segregation and cell cycle progression[33,35,36]. Interestingly, PP1 also regulates Par-3 function by dephosphorylating it and stabilizing its interaction with aPKC[37]. Therefore, we explored whether ASPP2-PP1 interaction was required in our experimental context. We rescued either wildtype ASPP2 (ASPP2-WTres) or a mutant ASPP2, in which the canonical RVXF motif essential for the binding to PP1 was mutated (KRVF to KARA; ASPP2-KARAres), in ASPP1,2 DKO cells. Fluorescence imaging and line scan analysis showed that both constructs were expressed to a similar level and equally enriched at apical junctions, although ASPP2-KARAres was somewhat more diffusely lateral ($P_{AJC} = 0.189$, $P_{non-AJC} < 0.001$; Fig. 4a and Supplementary Fig. S6a). Both ASPP2-WTres and ASPP2-KARAres restored Par-3 concentration at AJC, so that localized PP1 activity was dispensable to correctly localize Par-3 (Fig. 4a and Supplementary Fig. S6b–d). Strikingly, apical cell area remained highly irregular in ASPP2-KARAres cells, indicating still aberrant contractile activity in these cells ($P_{WTres} = 0.2692$, $P_{KARAres} = 0.0775$; Fig. 4b–e). Consistently, while ROCK1 was present at lateral and apical membranes, it was no longer enriched at AJC in ASPP2-WTres cells; by contrast, ROCK1 was strongly concentrated at AJC with activated α-catenin in ASPP2-KARAres cells (Fig. 4f, g).

Given these findings, we next considered the effect of ASPP depletion on aPKC localization. aPKC appeared to be enriched at cell junctions in WT cells but not in ASPP1,2 DKO cells (Supplementary Fig. S6e, f). However, ubiquitous non-junctional immunofluorescence at the apical membrane precluded clear examination by orthogonal projections. Therefore, we compared the 3D projections of the inset regions to elucidate the respective aPKC localizations. In this way, we identified that in WT cells, there was a distinctly junctional population that was highly colocalized with activated α-catenin, apart from the apical membrane population; by contrast, only the diffusely apical aPKC could be observed in ASPP1,2 DKO cells (Supplementary Fig. S6e). Likewise, application of the 3D projection analysis to ASPP2 rescue cells revealed that aPKC was strongly enriched at GFP-positive junctions in ASPP2-WTres cells but not in ASPP2-KARAres cells (Fig. 4h, i). Thus, ASPP2 interaction with PP1 is required to position aPKC to counteract ROCK-induced apical cell contractility.

**Normalization of intercellular tension is regulated by balancing junctional ROCK1 from cell to cell**. If apical contractility is uniformly upregulated throughout the epithelial cell sheet, we would expect to see smaller cells across the board and increased cell density as a result. However, we repeatedly found that the KO cell monolayer is a patchwork of cells with varying apical cell areas, which indicates that apical contractility is disproportionate from cell to cell. Therefore, we considered the complimentary possibilities that either ROCK1 or myosin—or both—could be accumulated unequally at AJC.

When we examined ROCK1 enrichment at AJC in each of 10 contiguous cells in WT, ZO-1,-2 DKO, and MAGI-1,-3 DKO monolayers, we found that there was a greater discrepancy in ROCK1 immunofluorescence in KO cells compared to WT cells. Notably, plotting ROCK1 intensity against their respective cell areas showed that WT cells clustered tightly, while KO cells were scattered, suggesting that nonuniformity of ROCK1 localization is a contributing factor in the irregularity of apical domains (Fig. 5a). Intriguingly, strength of ROCK1 immunofluorescence was inversely correlated with cell area (Fig. 5b).

Next, we examined myosin accumulation at AJC. Two isoforms of non-muscle myosin II, NMIIA and NMIIB, localize to the epithelial AJC[7]. While both isoforms are necessary for the mechanical integrity of epithelial junctions, NMIIB is considered the dominant generator of force at AJC and is required for sustaining high tension along the cell-cell interface[7,38]. Therefore, we examined NMIIB localization in our knockout cell models. Consistent with previous studies, we found that NMIIB colocalized with F-actin at cell junctions in WT cells and junctional enrichment was preserved in ZO-1,-2 DKO, MAGI-1,-3 DKO and ASPP1, 2 DKO cells (Fig. 5c). Intriguingly, NMIIB showed conspicuous juxtamembrane positioning alongside junctional F-actin in some cells of the KO cell sheets (arrowheads). It should be noted that in ZO-1,-2 DKO cells, NMIIB localization was not limited to the AJC and expanded to the subapical region, suggesting that ZO proteins regulate multiple signaling pathways beside the MAGI-ASPP-Par-3 axis that converge on apical contractility (brackets).

These results taken together indicate that homeostasis of the epithelial sheet morphology requires conformity of two factors within a certain range by the constituent cells: junctional ROCK1 activity and NMIIB recruitment. Finally, signaling from AJC to Par-3-aPKC through MAGI is a crucial means of normalizing this activity level, owing to MAGIs' role in regulating local phosphorylation dynamics through ASPP2-PP1 (Fig. 5d).

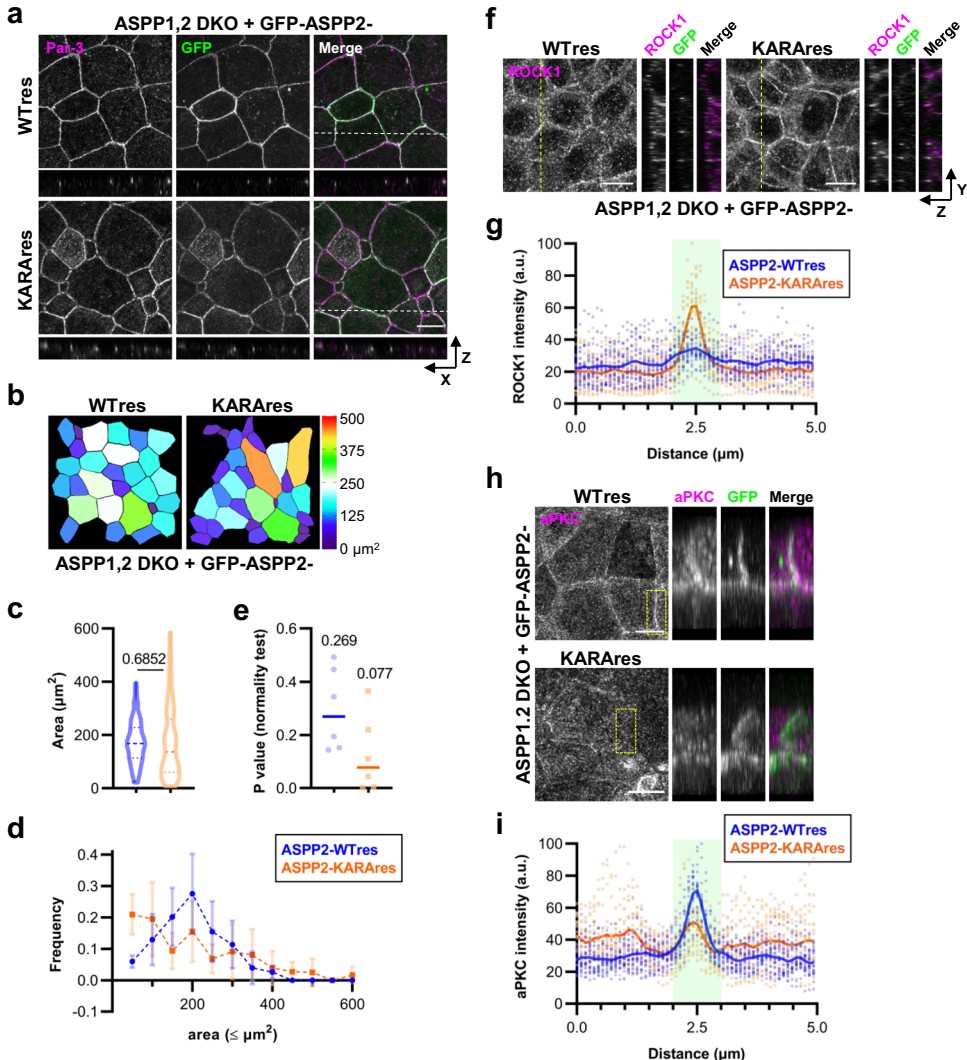

**Fig. 4 Par-3 recruitment to AJC is insufficient to downregulate ROCK in the absence of ASPP2-PP1 interaction. a** Representative immunofluorescence images of GFP-ASPP2-WTres and GFP-ASPP2-KARAres cells stained for Par-3 (magenta). Scale bar, 10 μm. **b** Pseudocolor representations of apical areas in GFP-ASPP2-WTres and GFP-ASPP2-KARAres cells. **c** Violin plot depiction of apical areas in GFP-ASPP2-WTres and GFP-ASPP2-KARAres cells. Data were collated from six measurements obtained over three independent experiments. Lines represent the median and the upper and lower quartiles. *P* value from unpaired *t* test is shown. *n* = 149 (WTres) and 139 (KARAres). **d** Frequency distributions of apical areas in GFP-ASPP2-WTres and GFP-ASPP2-KARAres cells. Data were as in **c**. Points are means and error bars are SDs. **e** P values of D'Agostino & Pearson test for normal distribution were computed from the data presented in **d**. Medians are notated and shown graphically as lines. **f** Representative immunofluorescence images of GFP-ASPP2-WTres and GFP-ASPP2-KARAres cells stained for ROCK1. Scale bar, 10 μm. **g** Cross-junctional line scans of ROCK1 immunofluorescence from images corresponding to **f**. Shaded area represents AJC as defined by GFP fluorescence. Individual data from 20 independent line scans are shown with the means depicted by solid lines. **h** Representative immunofluorescence images of GFP-ASPP2-WTres and GFP-ASPP2-KARAres cells stained for aPKC. 3D projections of the region indicated by the yellow dotted square are shown to the right. Scale bar, 10 μm. **i** Cross-junctional line scans of aPKC immunofluorescence from images corresponding to **h**. Shaded area represents AJC as defined by GFP fluorescence. Individual data from 20 independent line scans are shown with the means depicted by solid lines. Source data are available in Supplementary Data 1.

## Discussion

Based on our finding, we propose that apical tension is equivalently maintained among cells throughout the epithelial monolayer in a multistep process. First, cells must sense the tension through cell adhesion structures; then, it is necessary to modulate contractile activity at AJC within a steady-state range by regulating ROCK through some feedback mechanisms. Tensile force at AJC is sensed by mechano-sensitive proteins such as α-catenin[39,40]. However, it is unclear how such proteins convey the mechanical information to precisely modulate biochemical activity, e.g. RhoA-ROCK signaling, and alter cellular contractility. A double-negative feedback mechanism between RhoA-ROCK and Rnd3-p190 RhoGAP was proposed to confer robustness on junctional ROCK activity at the

population level in the epithelial cell sheet but it is not known how Rnd3 is regulated in this case[5].

In the present study, we revealed the means by which ROCK activity is downregulated at AJC; the scaffolding proteins, ZO and afadin, differentially recruit MAGI proteins, and by extension, a signaling complex centered around Par-3, to concentrate aPKC at apical junctions and delocalize ROCK. Conversely, ROCK is reported to disrupt Par complex formation by phosphorylating Par-3 and destabilizing its interaction with aPKC[41]. Therefore, mutual antagonism between Par-3/aPKC and ROCK might be involved in maintaining the homeostasis of intercellular tension. At present, it is unclear how this mechanism is mobilized and perpetuated. Interestingly, we previously showed that afadin,

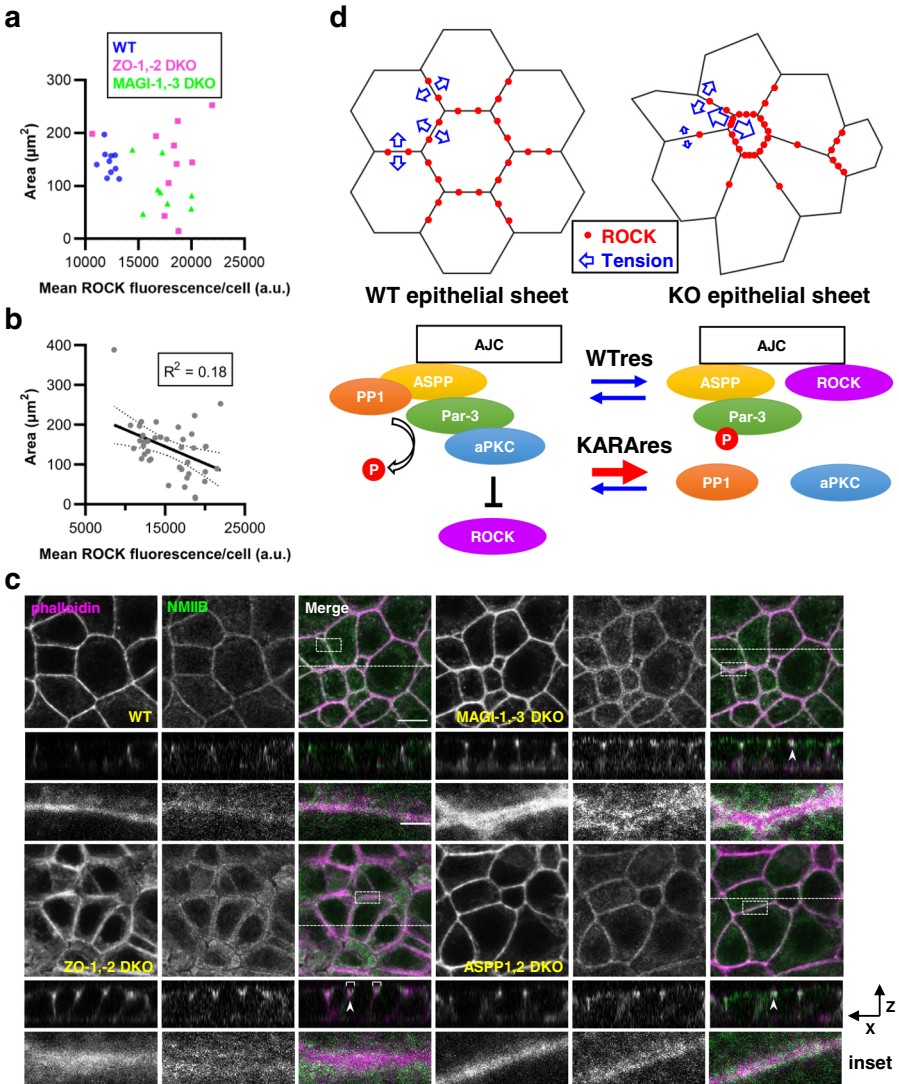

**Fig. 5 Antagonism of ROCK activity at AJC requires scaffolding of PP1-Par-3 interaction by ASPP2 to stabilize the Par-3-aPKC complex. a** Plot of cell area against ROCK1 immunofluorescence for WT, ZO-1,-2 DKO and MAGI-1,-3 DKO cells. Representative images were shown previously in Figs. 1h and 2f. Maximum intensity projections of apical sections determined by α-catenin enrichment were segmented. Data were from ten contiguous cells for each condition. **b** Plot of cell area against ROCK1 immunofluorescence from data in F collated with additional measurements. Solid line is the interpolated linear regression with the 95% confidence intervals shown by the dotted lines. **c** Representative immunofluorescence images of WT, ZO-1,-2 DKO, MAGI-1,3 DKO and ASPP1,2 DKO cells stained for NMIIB. Orthogonal views of the cross-section and insets indicated by the white dashed lines and boxes, respectively are shown below. Arrowheads show well-developed NMIIB filaments alongside the junctional actin network. Brackets show NMIIB expansion to the subapical region. Scale bar, 10 μm (inset, 2 μm). **d** Proposed model for normalization of intercellular tension in the epithelial cell sheet by MAGIs. Source data are available in Supplementary Data 1.

which is required to correctly localize MAGIs, is recruited to AJC by α-catenin in a tension-sensitive manner, highlighting how to force sensation could be incorporated in our proposed machinery[40]. Future studies will seek to clarify the functional network among these molecules to provide a comprehensive picture of the molecular mechanisms that maintain the integrity of epithelial cell sheets.

We previously found that there is a conspicuous disorganization of lymphatic vasculature in *Aspp1*-deficient mouse embryo that critically delays formation of the primary lymphatic network[42]. At the cellular level, *Aspp1*-deficient lymphatic endothelial cells showed morphological abnormalities suggestive of defects in cell adhesion and cytoskeletal rearrangement. In Drosophila, ASPP interaction with PP1 is essential for the proper organization of inter-ommatidial cells in the developing eye; intriguingly, functional interaction between ASPP and the

N-terminal RASSF protein, RASSF8, is also required[34]. These reports suggest that signaling from AJC to Par-3-aPKC through MAGIs is a highly conserved mechanism that is broadly applied during developmental morphogenesis. Further analyses in such experimental models could also shed light on the regulatory interaction between MAGIs and the AJC scaffolding proteins.

## Methods

**Cell culture and reagents**. Wildtype EpH4, EpH4 ZO-1,-2 DKO, EpH4 ZO-1,-2, afadin TKO, EpH4 ASPP1,2 DKO, EpH4 ASPP1,2 DKO rescue, EpH4 MAGI-1,-3 DKO, EpH4 GFP-RASSF10, and HEK293 cells were grown in DMEM supplemented with 10% FCS. We established KO EpH4 cells by using the CRISPR-Cas9 system. Oligonucleotides were phosphorylated, annealed, and cloned into the BsmBI site of pLenti-CRISPR v2 vector according to the Zhang laboratory protocols (F. Zhang, MIT, Cambridge, MA). The target sequences were as follows;
   Mouse Afadin: 5′- GAGGAGAGCATGCGCATGTC-3′
   Mouse ASPP1: 5′-GACTACAGCAAGATCATGAA-3′

Mouse ASPP2: 5′-GAGCAAGGAACCATCAGGCA-3′
Mouse MAGI-1: 5′-GGGGACCCCAGGGCGAGCTG-3′
Mouse MAGI-3: 5′-GAGGACCTGATCAACTGGAT-3′

The following primary antibodies were used for immunofluorescence (IF) microscopy, immunoprecipitation (IP) and immunoblotting (IB): rat anti-ZO-1 (DSHB; 1:50, IF; 1:500, IB); rabbit anti-ZO-2 (Zymed; 1:1000 IB); mouse anti-AF6 (afadin; BD Biosciences; 1:200, IF); rabbit anti-Par-3 (Merck Millipore; 1:300, IF); mouse anti-MAGI-1 (ss-5; 1:100, IF; 1:500, IB), mouse anti-MAGI-3 (46; 1:100, IF; 1:500, IB) and rabbit anti-PKCζ antibodies (Santa Cruz Biotechnology; 1:100, IF); mouse anti-phospho-myosin light chain 2 (Ser19; used in Supplementary Figs. S2H and S3I; 1:100, IF), rabbit anti-phospho-myosin light chain 2 (Ser19; used in Fig. 1C; 1:100, IF) and rabbit anti-Ezrin (CST; 1:100, IF); rabbit anti-ROCK1 (abcam; 1:200, IF); rabbit anti-Claudin-3 (Life Technologies; 1:100, IF); mouse anti-GFP (1:100, IF; 1:1000, IB) and rat anti-HA (Roche; 1:1000 IB); mouse anti-DYKDDDDK (FLAG; 1:1000 IB) and rat anti-E-cadherin (ECCD-2; Wako Pure Chemicals; 1:100, IF). Rat anti-activated α-catenin antibody (α18; 1:50, IF) was a kind gift from Dr. A. Nagafuchi (Nara Medical University, Nara, Japan). Rabbit anti-ASPP1 antibody (1:1,000 IB) and rabbit anti-ASPP2 antibody (1:100, IF; 1:1000, IB) were generated by Dr. M. Hirashima.

**Immunofluorescence microscopy and image processing**. Cells cultured on coverslips were fixed with 3% formalin prepared in PBS for 15 min at room temperature (RT), permeabilized with 0.4% Triton X-100/PBS for 5 min and blocked with 1% BSA prepared in PBS for 30 min at RT. Antibodies were diluted in the blocking solution. Cells were incubated with primary antibodies for 1 h at RT and with secondary antibodies for 45 min at RT. For cell size measurement, cells were cultured to confluence and incubated with 10 μM ROCK inhibitor (Y-27632; Sigma) or an equivalent volume of the vehicle (DMSO) for 5 h at 37 °C.

Immunofluorescence images were obtained using a 63× oil-immersion objective on an inverted microscope interfaced to a laser-scanning confocal unit (LSM700; ZEISS). Images were acquired using ZEN2012 software and processed using ImageJ/Fiji. Images presented are maximum intensity projections, orthogonal projections, and 3D projections.

Cross junctional line scan analysis was performed on maximum intensity projection images. Line segments (5 μm) were drawn perpendicular to cell junctions, centered around the reference immunofluorescence (activated α-catenin etc). Line widths were set to pixel values approximately equal to 2 μm so that the obtained fluorescence intensity values were means over the width of the line.

Cells were segmented based on maximum projection images of activated α-catenin immunofluorescence to define "Cell junction" (AJC) and "Cytosol" (cell area) using the Trainable Weka Segmentation plugin according to the schematic in Supplementary Fig. S1c[43]. Cell areas were obtained by the "Analyze Particles…" command excluding objects on edges and representative images were annotated in pseudocolor using the ROI Color Coder plugin. Immunofluorescence relative to AJC were analyzed using the "Image Calculator…" command to multiply the segmented AJC mask over the immunofluorescence image of interest, thresholding the image to remove non-AJC background and extracting "Mean gray value" and "Area" data using the 'Measure' command. Junctional coverage was defined as rate of immunofluorescence area per total AJC area.

**Immunoblotting**. Cells were cultured to confluence, lysed in 2 × SDS sample buffer and boiled for 5 min. Samples were resolved by SDS-PAGE and transferred to nitrocellulose membrane. The membrane was blocked in 5% skim milk/0.1% Tween 20/TBS for 30 min at RT. Indicated proteins were probed by sequential incubation with the primary antibody and HRP-conjugated secondary antibody prepared in blocking buffer for 1 h at RT each. The membrane was exposed to SuperSignal West Dura detection reagent (Thermo Fisher) and chemiluminescence was captured using the LAS-3000 Imaging system (Fujifilm).

**FLAG immunoprecipitation for heterologous protein interaction**. Cells were washed with ice-cold PBS and lysed with FLAG IP buffer (20 mM Tris-HCl [pH 7.5], 150 mM NaCl, 1% Triton X-100, and protease inhibitors). Lysates were incubated with 5 μl of anti-DYKDDDDK mAb beads (Wako Pure Chemical Industries) for 2 h. Beads were washed with FLAG IP buffer and bound proteins were eluted in SDS sample buffer. Aliquots of the lysate and eluate were immunoblotted with anti- DYKDDDDK, anti-HA and anti-GFP antibodies.

**Immunoprecipitation for endogenous protein interaction**. Cells were washed with ice-cold PBS and lysed in endogenous IP buffer (20 mM HEPES-NaOH, pH 7.4, 100 mM NaCl, 5 mM EDTA, 10% glycerol, 1% Triton X-100, and protease inhibitors). Lysates were incubated with either isotype control or specific antibodies (2–8 μg) overnight in the cold with end-over-end mixing. Protein G–Sepharose (GE Healthcare Life Sciences) was added to the lysates, and incubation was extended for 1.5 h. The beads were washed with endogenous IP buffer and dissolved in SDS sample buffer.

**Statistics and Reproducibility**. GraphPad Prism 8.4.1 (GraphPad Software) was used to graph data and to perform statistical analyses. One-way ANOVA, unpaired t test or multiple t tests was performed as appropriate to compare means.

Normality of data distribution was assessed by D'Agostino & Pearson's test. Error bars are SD and P values are notated in the figures.

Line scan analyses were performed on pooled data across three independent experiments. Images for apical area quantification were obtained over three independent experiments and two fields of view per each experiment. Areas were collated; sample sizes are provided in the legend. Immunoblots and immunofluorescence images are representative of at least three independent experiments.

**Reporting summary**. Further information on research design is available in the Nature Research Reporting Summary linked to this article.

## Data availability
All data supporting the findings of this study are included in the paper and its supplementary information files. Source data for the graphs as well as the processed microscopy images underlying the cell area analyses are provided as Supplementary Data 1. All relevant data are available from the corresponding author on reasonable request.

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

## Acknowledgements

We thank all members of the Ikenouchi laboratory (Department of Biology, Faculty of Sciences, Kyushu University) for helpful discussions. This work was supported by MEXT/JSPS KAKENHI (JP19H04968 (J.I.), JP19H03227 (J.I.), JP19K06640 (K.M.), JP24112513 (M.H.), and JP26460253 (M.H.)), AMED-PRIME (15664862) and grants from the MSD Life Science Foundation and the Sumitomo Foundation.

## Author contributions

K.M. performed most of the experiments, analyzed the data and wrote the paper. H.O., K.S., and T.S. performed experiments. H.M. generated anti-ASPP1 and anti-ASPP2 antibodies. J.I. designed research and wrote the paper.

## Competing interests

The authors declare no competing interests.
