## [Peer Review File · Communications Biology]

Reviewers' comments:

Reviewer #1 (Remarks to the Author):

The authors started this study with the observation of irregular apical morphology of ZO1/2-KO cells. ROCK is concentrated at apical junctional complexes (AJC) and ROCK inhibition recovers the morphology, supporting that ROCK activity contributes to the irregular morphology. Consistently, Par-3, which suppresses ROCK localization at AJC, is mislocalized from AJC in ZO1/2-KO cells. Thereafter the authors revealed the following findings step by step. i) ZO1/2 and afadin are involved in the accumulation of MAGI1/3 at AJC; ii) MAGI1/3-KO cells also exhibit irregular apical morphology, which is recovered by ROCK inhibition; iii) Par-3 is not properly localized at AJC in MAGI1/3-KO cells; iv) MAGI3, ASPP2, and RASSF10 form a complex; v) ASPP2 is reduced at AJC in MAGI1/3-KO cells, while MAGI1/3 localization is not affected by ASPP1/2-KO; vi) ROCK1 is enriched and Par-3 is reduced at AJC in ASPP1/2-KO cells, which show irregular apical morphology; v) ASPP2 lacking PP1A-interacting region fails to rescue the localization of Par-3 and aPKC at AJC in ASPP1/2-KO cells, suggesting that ASPP1/2 stabilizes Par-3/aPKC interaction through dephosphorylation of Par-3: and iv) ROCK distribution is diverse in ZO1/2-KO and MAGI1/2-KO cells. Based on these findings the authors concluded that ZO1/2 (and afadin) determine the localization of MAGI1/2 at AJC, which recruit APSS1/2, induce dephosphorylation of Par-3 through PP1A to stabilize Par-3/aPKC complex, so that Par-3/aPKC are accumulated at AJC and ROCK activity is restricted. The story may be interesting for researchers who study cell junctions and cell polarity. Overall, the experiments are logically designed but at some parts, there are gaps.

Major comments:

- 1: Unfortunately, the findings to support the proposed interactions (ZO1/MAGI, Afadin/MAGI, MAGI3/RASSF10, ASPP2/RASSF10) among endogenous proteins are missing.
- 2: The model that ASPP2 promotes dephosphorylation of Par-3 and strengthens the interaction between Par-3 and aPKC is attractive. However, there is no evidence that Par-3 dephosphorylation is reduced in ASPP1/2-KO cells and that Par-3 and aPKC more tightly interact with each other.
- 3: This might be out of scope of this study. However, readers may ask what finally determines the apical morphology. In this point of view, it would be desirable to demonstrate myosin II in ZO1/2-KO, MAGI1/3-KO, and ASPP1/2-KO cells.

Minor comments:

- 1: There are several grammatical errors. The authors should use a certain authorized editing service.
- 2: Better to add one figure to show the proposed model.
- 3: The authors describe "It was previously shown that Par-3 localization at apical junctions is disrupted in MDCK cells depleted of ASPP2 but the precise molecular mechanisms remained unclear." However, in Ref 24, the researchers demonstrated the interaction between endogenous Par-3 and ASPP2. It would be better to describe this interaction. Otherwise, readers unfamiliar with this topic cannot figure out how Par-3 is recruited to AJC. In my understanding of this manuscript, the hierarchy of molecular interactions in AJC complex is supposed to be ZO1/2/Afadin-MAGI1/3-ASPP1/2-Par-3-aPKC-ROCK.
- 4: Could you explain more closely what you did in Figure S1?

Reviewer #2 (Remarks to the Author):

Rationale for study: Epithelial cells are coupled with each other through an apical junction complex of actin-rich Tight and Adherens Junction constituents, where RhoA/ROCK signaling coordinates the

contractility/ coupling of actin and myosin filaments adjacent to these junctions. Evidence from previous KO and OE studies show requirements of individual components for Rho/ROCK signaling at apical junctions (AJs), but the hierarchy of molecular interactions that drive/balance apical adhesions are only just coming into view, particularly regarding how Rho/ROCK and Par3/aPKC mutually antagonistic signaling plays a role.

Findings: This study adds information on new protein-protein module relationships required to balance intercellular tension of apical adhesions. While the study may not tell a full coherent story, it nevertheless shows that KO/loss of a number of junctional components lead to elevated ROCK activity (ZO proteins, MAGIs 1&3, ASPP1/2), through their inability promote the uniform distribution of Par3 at AJs (where Par3/aPKC is known to limit ROCK activity at AJs).

Novelty of the study is three-fold: 1) Authors confirm previous work (Peifer lab) that loss of ZO proteins dysregulates ROCK-dependent contractility to alter apical morphology using a different cell system (Eph4 mouse mammary gland cells, an established immortalized line). The authors do this by quantifying the distribution of apical areas across the monolayer, which reflect degree to which AJC is under tension using a segmentation protocol (Fig. S1). Thus, the authors define a new way to quantify consequences of unbalanced contractile forces on cells. 2) Authors show MAGI proteins antagonize apical constriction, similar to what has been observed with known MAGI-binding partners (ZO-1/ZO-2, afadin). They show that MAGI is required to promote the uniform distribution of Par-3 along apical junctions, where Par3/aPKC are known to limit ROCK activity at AJs. 3) Authors show MAGI forms a ternary complex with the Par3 regulator, ASPP and N-terminal Rassf proteins and map these interactions (MAGI and the PDZ-interaction motif of Rassf10, and that the coil-coil N2 domain of Rassf10 is required and sufficient to bind the Par3-interactor ASPP). 4) An ASPP-PP1 interaction (making use of ASPP2-mutant that fails to bind PP1) is required for ASPP scaffolding of PP1 and Par3, required to stabilize Par3/aPKC and limit ROCK activity.

Assessment: Overall, this study provides important new information on protein-protein interactions at AJs that control ROCK-recruitment/balanced adhesions. Figures are largely clear, quantifications robust.

Suggestions Minor:

1. All fluorescence quantification graphs should have the y-axes labelled for the protein intensity measurement of interest. It would make the figures easier to follow.
2. Fig. 5D, F and G would benefit from showing transfection with GFP as control. This is seen in the blot but confusing when not specified.

Reviewer #3 (Remarks to the Author):

In this manuscript, Matsuzawa et al. use elegant co-cultures of wild-type with Crispr generated knock-out Eph4 mouse mammary cells, and confocal microscopy and image segmentation approaches to study the role of several apical scaffolds in the regulation of apical cell size and shape. They describe that ZO1&2, MAGI1&3, and ASPP1&2 are apical scaffolds that negatively regulate ROCK1 activity and pMyosin junctional accumulation thus helping homogenise tension across the epithelial sheet ensuring homogeneity of cell sizes. Without these different scaffolds, cells apices exhibit great size diversity. Using Co-IP experiments the authors then describe new protein interactions between MAGI3 and ZO1&2 and with Afadin. They also show that similarly to its Drosophila homologues, MAGI3 interacts with RASSF10 and ASPP2. Using rescue experiment, they then study the role of the PP1 interaction

domain of ASPP and describe that even though the PP1 interaction is not required for ASPP-mediated Par3 localisation, it is required for ROCK1 inhibition, cell apex sizes regulation, and the accumulation of the Par3 partner aPKC at apical membranes. Given the previously reported role of aPKC as an inhibitor of ROCK1, the authors propose a model by which ZO recruit MAGI/RASSF/ASPP to localise aPKC inhibiting ROCK1 and thus homogenising tensile forces across the epithelial sheet.

The description of the cell size phenotypes and of the mislocalisations of the different scaffolds and ROCK1/Myosin is very nicely performed and the insights gained here should be of interest to the field. In particular it is extending previous work done in *C. elegans* and in *Drosophila*, bringing new developments to the role of MAGI scaffolds and the associated RASSF/ASPP complexes during junctions and actin dynamics, and bringing interesting hypotheses. There are however several points that are not entirely supported by the data presented and they would need to be either further experimentally supported or to be weakened in the text before I can support this study for publication. Below are detailed the major points that should be clarified

1. The relations between ZO / MAGI / ASPP

The role of these different scaffolds is studied one at a time. They all produce the same phenotype: cell size heterogeneity, ROCK1 hyperactivation, pMyosin increase, and lower Par3 recruitment. However, the paper concludes or implies that given similar phenotypes they should be in a same pathway/complex... This is to my view an overinterpretation of the data. I acknowledge that there is biochemistry to show mutual binding (more on that in point 2.) but this is not enough to say they that for instance MAGI mediate the effects of ZO. Authors should either

- i) perform rescue experiments in which the interaction domains are deleted (similarly to what they did for ASPP2-KARAs) and show that preventing the interaction prevents rescue
- ii) change significantly the text to acknowledge that more work is required to formally prove these different scaffolds work together.

Indeed an alternative mechanism that is not explored at all by authors is that ZO /Afadin/ MAGI / ASPP... work in parallel pathways all ultimately converging towards ROCK1 inhibition. This point should be discussed.

2. Biochemistry and Co-IPs

The authors identify that MAGI-1 and MAGI-3 differ in their response to ZO-1&2 loss: MAGI-3 is almost completely delocalized, while substantial MAGI-1 staining remained. Then the authors test binding between MAGI-1 or MAGI-3 with Afadin and ZO-1 (Fig. 3). Several controls are missing before reaching their conclusions. First, authors should also test ZO-2. Indeed part of their reasoning is that MAGI-3 binds more to ZO-1 and hence this is why it is more strongly delocalised. One could argue that MAGI-1 binds more to ZO-2 and the reasoning would fall apart...

Second, in panel 3E, I fail to see the ZO-1 input?

Third, the authors should show the effect of single Afadin KO. Indeed, Page 6 lines 10-12, they argue that Par3 is mislocalised in triple Afadin / ZO-1 / ZO-2, but not in ZO-1/ZO-2 double, thus that the remaining Par3 in ZO-1 / ZO-2 double is because of the remaining MAGI-1... This statement is too strong for observations that are mainly correlative. Par3 could simply be localised by Afadin in their set-up.

3. Role of PAR3 and aPKC?

The paper ends with a deeper analysis of the role of ASPP2 and in particular with respect to the binding of ASPP2 with PP1. They show that the interaction between ASPP and PP1 is not required for Par3 localisation at membranes, but required for cell size homogeneity and ROCK1 inhibition. They then show that aPKC is mislocalised, thus proposing that aPKC stabilisation mediates ROCK1 inhibition (building on previous literature). This observation is intriguing since the stabilisation of aPKC has been shown to be dependent of Par3. Despite, Par3 still being well localised, the authors continue including

Par3 in their model. I might have missed something, but I do not really understand why. Authors should discuss how they view the decoupling of Par3 and aPKC in this context?

Minor remarks

- I think panel 4B and 4C are miscalled in the text
- Fig. 6A&B: MAGI-1 seems higher in ASPP1&2 KO cells unlike what is claimed in the text
- Could the authors explain a bit more the EpH4 culture conditions. Are cells allowed to reach full confluency and then imaged? Is the phenotype of cell size disparity seen early or just at a certain culture stage?
- Interestingly, in the tension dispersion model proposed, I would have anticipated that cell size heterogeneities would be transmitted, at least slightly, to wild-type neighbouring cells... But this is never observed in the pictures provided. Maybe authors would like to add a sentence explaining why the phenotype is so strictly cell autonomous.

We are grateful for the constructive criticism from all the Reviewers. In order to better focus on our principal findings, we have reorganized the figures between main and supplementary data. Below are the point-by-point responses to each of the Reviewers' concerns:

Reviewer #1

The story may be interesting for researchers who study cell junctions and cell polarity. Overall, the experiments are logically designed but at some parts, there are gaps.

Comment 1-1

Unfortunately, the findings to support the proposed interactions (ZO1/MAGI, Afadin/MAGI, MAGI3/RASSF10, ASPP2/RASSF10) among endogenous proteins are missing.

Response 1-1

We performed immunoprecipitation assays to examine interactions among endogenous proteins. Both MAGI-1 and MAGI-3 co-precipitated with ZO-1 (and afadin) and we were able to detect MAGI-1 with the afadin immunoprecipitate. Several commercial antibodies against RASSF10 failed to detect reliably endogenous level of protein by immunoblotting. Therefore, we stably expressed FLAG-tagged RASSF10 in Eph4 cells for the purpose of biochemical analysis. We found that FLAG-RASSF10 was coprecipitated with both MAGI-1 and MAGI-3 and that ASPP2 coprecipitated with FLAG-RASSF10. These results are shown in **Supplementary Fig. S2h, i** (afadin/ZO-1-MAGI), **Supplementary Fig. F4e, f** (MAGI-RASSF10) and **Supplementary Fig. S4g** (RASSF10-ASPP2) of the revised manuscript.

Comment 1-2

The model that ASPP2 promotes dephosphorylation of Par-3 and strengthens the interaction between Par-3 and aPKC is attractive. However, there is no evidence that Par-3 dephosphorylation is reduced in ASPP1/2-KO cells and that Par-3 and aPKC more tightly interact with each other.

Response 1-2

We analyzed Par-3 phosphorylation and Par-3-aPKC interaction in WT and ASPP1,2 DKO Eph4 cells by immunoprecipitating Par-3. Immunoblotting with an anti-phosphoSer/phosphoThr antibody showed that bulk serine/threonine phosphorylation of Par-3 was largely unaltered between WT and ASPP1,2 DKO cells (**Fig. R1**). Immunoblotting with a phosphorylation-specific antibody to Par3-Ser144, which is reportedly phosphorylated by ROCK in vivo (Nishioka et al. 2012. Cell Struct Funct), also did not reveal a difference between WT and ASPP1,2 DKO cells (**Fig. R1**). We also found that aPKC co-precipitated with Par-3 with equal efficiency from WT and ASPP1,2 DKO cells.

Regulation of Par-3 function by phosphorylation is highly complex and often depends on

subcellular context. Besides Ser144, Thr833 is also phosphorylated by ROCK; phosphorylation of Thr833 as well as that of Ser827 and Ser829 negatively regulate Par-3-aPKC binding (Nagai-Tamai et al. 2003. *Genes Cells*; Nakayama et al. 2008. *Dev Cell*). More recently, Yamashita et al. reported that Ser852 and Ser889 are susceptible to dephosphorylation by PP1 (2020. *J Cell Sci*).

Par-3 interaction with aPKC is essential for apical domain development (Horikoshi et al. 2009. *J Cell Sci*). Our observations show that apical-basal polarization is normal in MAGI-1,-3 DKO (**Supplementary Fig. S3c**) and ASPP1,2 DKO cells, suggesting that there is a significant pool of non-junctional Par-3 and aPKC in complex at steady state. This could explain why we could not detect a difference in Par-3-aPKC complex formation between WT and ASPP1,2 DKO cells, since immunoprecipitation can only detect bulk interactions. Given these limitations, we believe that our model is consistent with the cell biological data, which aligns with previous studies regarding the mechanism of ROCK antagonism by aPKC. Resolving the phosphorylation states of Par-3 and how Par-3 interacts with aPKC in the context of junctional force distribution are important undertakings that require considerable effort beyond the scope of this study. We look forward to tackling these questions in future works.

Fig. R1 Par-3 was immunoprecipitated from WT and ASPP1,2 DKO EpH4 cells and analyzed by immunoblotting with the indicated antibodies.

Comment 1-3

This might be out of scope of this study. However, readers may ask what finally determines the apical morphology. In this point of view, it would be desirable to demonstrate myosin II in ZO1/2-KO, MAGI1/3-KO, and ASPP1/2-KO cells.

Response 1-3

We examined the localization of non-muscle myosin IIB in WT and the various KO cells. The results are presented as **Fig. 5c**, with the description amended in the revised text as follows:

Page 11 Line 11

Next, we examined myosin accumulation at AJC. Two isoforms of non-muscle myosin II, NMIIA and NMIIB, localize to the epithelial AJC7. While both isoforms are required for the mechanical integrity of epithelial junctions, NMIIB is considered the dominant generator of force at AJC and is required

for sustaining high tension along the cell-cell interface. Therefore, we examined NMIIB localization in our knockout cell models. Consistent with previous studies, we found that NMIIB colocalized with F-actin at cell junctions in WT cells and junctional enrichment was preserved in ZO-1,-2 DKO, MAGI-1,-3 DKO and ASPP1,2 DKO cells (Fig. 5c). Intriguingly, NMIIB showed conspicuous juxtamembrane positioning alongside junctional F-actin in some cells of the KO cell sheets (arrowheads). It should be noted that in ZO-1,-2 DKO cells, NMIIB localization was not limited to the AJC and expanded to the subapical region, suggesting that ZO proteins regulate multiple signaling pathways beside the MAGI-ASPP-Par-3 axis that converge on apical contractility (brackets).

Comment 1-4

There are several grammatical errors. The authors should use a certain authorized editing service.

Response 1-4

The revised manuscript was reviewed by an editing service.

Comment 1-5

Better to add one figure to show the proposed model.

Response 1-5

We have updated our model (**Fig. 5d**) to specify the molecular mechanism involved in normalizing junctional tension.

Comment 1-6

The authors describe “It was previously shown that Par-3 localization at apical junctions is disrupted in MDCK cells depleted of ASPP2 but the precise molecular mechanisms remained unclear.” However, in Ref 24, the researchers demonstrated the interaction between endogenous Par-3 and ASPP2. It would be better to describe this interaction. Otherwise, readers unfamiliar with this topic cannot figure out how Par-3 is recruited to AJC. In my understanding of this manuscript, the hierarchy of molecular interactions in AJC complex is supposed to be ZO1/2/Afadin-MAGI1/3-ASPP1/2-Par-3-aPKC-ROCK.

Response 1-6

We have amended the text as follows with the altered text shown in red:

Page 7 Line 30

How do MAGI proteins control the targeting of Par-3 to AJC? **One intriguing candidate is ASPP2, which was shown to directly bind Par-3 and the loss of which disrupts hat Par-3 localization at apical**

junctions in MDCK cells. However, it is unknown how ASPP2 is recruited to AJC. ASPP2 colocalized with MAGI-3 at AJC in WT cells.

Comment 1-7

Could you explain more closely what you did in Figure S1?

Response 1-7

We performed pixel-based segmentation to define AJC based on immunofluorescence images in order to automate image quantification. Supplementary Fig. S1c shows the training workflow implemented in the Trainable Weka Segmentation plugin for ImageJ/Fiji. Representative images were processed to reduce noise and enhance edges by applying the indicated filters. Image pixels were then manually classified as either 'cell junction' or 'cytosol' to train and refine the protocol until automated segmentation images for new input images adequately delineated cell junctions. The probability images thus obtained were binarized and dilated to eliminate discontinuous cell junctions. The output images were used either to obtain apical cell area or to quantify AJC enrichment of proteins of interest, e.g. **Fig. 1k**. The relevant parts of the above description were added to the revised manuscript in the legend accompanying **Supplementary Fig. S1c**.

Reviewer #2

Overall, this study provides important new information on protein-protein interactions at AJs that control ROCK-recruitment/balanced adhesions. Figures are largely clear, quantifications robust.

Comment 2-1

All fluorescence quantification graphs should have the y-axes labelled for the protein intensity measurement of interest. It would make the figures easier to follow.

Response 2-1

We have labelled the y-axes of fluorescent quantification graphs with the protein intensity being measured.

Comment 2-2

Fig. 5D, F and G would benefit from showing transfection with GFP as control. This is seen in the blot but confusing when not specified.

Response 2-2

GFP used as control is specifically indicated in the revised figures (**Supplementary Fig. S2e-g, S4b-**

d) according to the reviewer's suggestion.

Reviewer #3

The description of the cell size phenotypes and of the mislocalisations of the different scaffolds and ROCK1/Myosin is very nicely performed and the insights gained here should be of interest to the field. There are however several points that are not entirely supported by the data presented and they would need to be either further experimentally supported or to be weakened in the text before I can support this study for publication.

Comment 3-1

1. The relations between ZO / MAGI / ASPP

The role of these different scaffolds is studied one at a time. They all produce the same phenotype: cell size heterogeneity, ROCK1 hyperactivation, pMyosin increase, and lower Par3 recruitment. However, the paper concludes or implies that given similar phenotypes they should be in a same pathway/complex... This is to my view an overinterpretation of the data. I acknowledge that there is biochemistry to show mutual binding (more on that in point 2.) but this is not enough to say they that for instance MAGI mediate the effects of ZO. Authors should either

i) perform rescue experiments in which the interaction domains are deleted (similarly to what they did for ASPP2-KARAs) and show that preventing the interaction prevents rescue

) change significantly the text to acknowledge that more work is required to formally prove these different scaffolds work together.

Indeed an alternative mechanism that is not explored at all by authors is that ZO /Afadin/ MAGI / ASPP... work in parallel pathways all ultimately converging towards ROCK1 inhibition. This point should be discussed.

Response 3-1

We attempted to narrow down the MAGI regions necessary for interacting with ZO-1, afadin and RASSF10 by performing a binding assay using MAGI-3 fragments. The fragments were as follows: fragment 1, N-terminal to GUK; fragment 2, middle PDZ domains; fragment 3, remaining C-terminal region. We found that ZO-1 bound to fragment 3, that afadin bound to fragment 1 and that RASSF10 bound to fragment 2. Therefore we performed rescue experiments in MAGI-1,-3 DKO cells using FL and deletion mutants (Δ COOH, defective in ZO-1 but not afadin binding; Δ PDZs, defective in RASSF10 binding) and examined the localization of ROCK1. All rescue constructs localized to cell junctions as expected (**Fig. R2**). However, whereas ROCK1 was delocalized from AJC in FL and Δ COOH rescue cells, it remained enriched at AJC in Δ PDZs rescue cells, the only one defective for

interacting with the putative downstream target, RASSF10(-ASPP2). These results suggest that MAGI, indeed, mediate the effect of ZO proteins and afadin to regulate junctional contractility. Given the complexity of these interactions, we feel that addressing the mode of MAGI interactions with upstream (AJC scaffolds) and downstream (RASSF, ASPP) interactors is best served as a separate research topic as our preliminary findings only scratch the surface. In further support of our proposed signaling axis, we now show interactions among endogenous proteins in the revised manuscript as **Supplementary Fig. S2h,i** (afadin/ZO-1-MAGI), **Supplementary Fig. F4e,f** (MAGI-RASSF10) and **Supplementary Fig. S4g** (RASSF10-ASPP2).

Fig. R2 a Schematic of the GFP-MAGI-3 rescue constructs and the fragments used to narrow down the binding region for afadin (**b**), ZO-1 (**c**) and RASSF10 (**d**). The N-terminal Fragment 1 bound afadin, the middle Fragment 2 containing multiple PDZ domains bound RASSF10 and the C-terminal Fragment 3 bound ZO-1. (**c**) Immunofluorescence images of ROCK1 staining in GFP-MAGI-3 rescue cells. Δ PDZs lacks the RASSF10-binding region and Δ COOH lacks the ZO-1-binding region.

However, we share the reviewer's misgiving that alternative mechanisms exist, and it was not our intention to propose a singular, linear signaling module. For example, ZO proteins have been shown to negatively regulate the junctional accumulation of Shroom3, an activator of ROCK1 (Choi et al. 2016. J Cell Biol). ZO proteins and afadin also regulate junctional contractility by accumulating

RhoGEFs and stabilizing adhesion anchoring to the circumferential actomyosin network (Nakajima and Tanoue 2012. Small GTPases; Matsuzawa et al. 2018. Cell Rep). In addition, further experiments in our KO cells showed that the junctional organization of the non-muscle myosin II isoform NMIIIB was clearly different in ZO-1,-2 KO cells compared to MAGI-1,-3 KO and ASPP1,2 KO cells (**Fig. 5c**). This is discussed further in our response to **Reviewer #1 (Response 1-3)** and at several points in the revised manuscript (**Page 7 Line 22; Page 11 Line 21**).

Comment 3-2

2. Biochemistry and Co-IPs

The authors identify that MAGI-1 and MAGI-3 differ in their response to ZO-1&2 loss: MAGI-3 is almost completely delocalized, while substantial MAGI-1 staining remained. Then the authors test binding between MAGI-1 or MAGI-3 with Afadin and ZO-1 (Fig. 3). Several controls are missing before reaching their conclusions. First, authors should also test ZO-2. Indeed part of their reasoning is that MAGI-3 binds more to ZO-1 and hence this is why it is more strongly delocalised. One could argue that MAGI-1 binds more to ZO-2 and the reasoning would fall apart...

Response 3-2

We examined the interaction between MAGI and ZO-2. Although MAGI-1 did not bind more to ZO-2, as proposed by the reviewer, we found that both MAGI-1 and MAGI-3 bound to ZO-2 with equal apparent affinity, in contrast to the interaction with ZO-1. This further highlights the complexity of MAGI interactions with their binding partners, which we hope to address in future works. The results are presented as **Supplementary Fig. S2g**.

Comment 3-3

Second, in panel 3E, I fail to see the ZO-1 input?

Response 3-3

The figures for MAGI-afadin (**Supplementary Fig. S2e**) and MAGI-ZO-1 (**Supplementary Fig. S2f**) interactions have been replaced to clearly show the input.

Comment 3-4

Third, the authors should show the effect of single Afadin KO. Indeed, Page 6 lines 10-12, they argue that Par3 is mislocalised in triple Afadin / ZO-1 / ZO-2, but not in ZO-1/ZO-2 double, thus that the remaining Par3 in ZO-1 / ZO-2 double is because of the remaining MAGI-1... This statement is too strong for observations that are mainly correlative. Par3 could simply be localised by Afadin in their set-up.

Response 3-4

As the reviewer notes, one possible interpretation is that afadin itself localizes Par-3 to AJC. We tested this possibility by examining Par-3 localization in afadin KO cells. Par-3 was equally enriched at AJC of afadin KO cells and WT cells, indicating that ZO proteins are sufficient to localize Par-3 (**Supplementary Fig S2d**). The revised text is amended with the following description (added text in red):

Page 6 Line 12

Crucially, Par-3 no longer accumulated at cell-cell junctions in ZO,-1,-2, afadin TKO cells (Supplementary Fig. S2c). **One possible interpretation here is that afadin recruits Par-3 to AJC. However, examination of Par-3 localization in afadin KO cells showed that Par-3 enrichment to AJC was unaltered in the absence of afadin (Supplementary Fig. S2d),** suggesting that junctional Par-3 in ZO-1,-2 DKO cells were recruited by remnant MAGI-1.

Comment 3-5

3. Role of PAR3 and aPKC?

The paper ends with a deeper analysis of the role of ASPP2 and in particular with respect to the binding of ASPP2 with PP1. They show that the interaction between ASPP and PP1 is not required for Par3 localisation at membranes, but required for cell size homogeneity and ROCK1 inhibition. They then show that aPKC is mislocalised, thus proposing that aPKC stabilisation mediates ROCK1 inhibition (building on previous literature). This observation is intriguing since the stabilisation of aPKC has been shown to be dependent of Par3. Despite, Par3 still being well localised, the authors continue including Par3 in their model. I might have missed something, but I do not really understand why. Authors should discuss how they view the decoupling of Par3 and aPKC in this context?

Response 3-5

A previous study showed that Par-3 is recruited to apical cell junctions through direct binding with ASPP2 (Cong et al. 2010. Curr Biol). Thus, it was to be expected that Par-3 localization was restored even in the absence of ASPP2-PP1 interaction. The reviewer is correct to point out that aPKC stabilization at cell junctions is dependent on Par-3 but it has also been shown that Par-3-aPKC interaction is sensitive to Par-3 phosphorylation and that PP1 can dephosphorylate Par-3 to enhance complex formation (**Page 9 Line 31**; Nagai-Tamai et al. 2002. Genes Cells; Traweger et al. 2008. PNAS). We believe that these studies offer a logical backdrop to interpret our observations and have proposed a model whereby ASPP2-PP1 stabilizes Par-3-aPKC interaction at AJC by modulating the phosphorylation state of Par-3. This may have been ambiguous in our initial manuscript and we have

added a schematic of the molecular mechanism to the model shown in **Fig. 5d**. Related issues are also discussed in **Response 1-2** to **Reviewer #1**'s comment.

Comment 3-6

I think panel 4B and 4C are miscalled in the text

Response 3-6

The text referencing the figures (**Supplementary Fig. S2b,c**) was corrected in the revised text.

Comment 3-7

Fig. 6A&B: MAGI-1 seems higher in ASPP1&2 KO cells unlike what is claimed in the text.

Response 3-7

We have replaced the images in question with more representative ones.

Comment 3-8

Could the authors explain a bit more the EpH4 culture conditions. Are cells allowed to reach full confluency and then imaged? Is the phenotype of cell size disparity seen early or just at a certain culture stage?

Response 3-8

Cells were allowed to reach confluence and cultured for an additional day to enable maturation of cell adhesions and apical-basal polarization. The phenotype of cell size disparity was most conspicuous in high cell density, established epithelial monolayers.

Comment 3-9

Interestingly, in the tension dispersion model proposed, I would have anticipated that cell size heterogeneities would be transmitted, at least slightly, to wild-type neighbouring cells... But this is never observed in the pictures provided. Maybe authors would like to add a sentence explaining why the phenotype is so strictly cell autonomous.

Response 3-9

The images of WT and KO co-cultured cells are limited to a few cells in the vicinity of the WT-KO boundaries. Therefore, it is insufficient to clearly depict the transmission of cell size heterogeneity from KO cells to WT cells. We include here a representative, expanded image of the co-cultured cell sheet (**Fig. R3**). WT cells immediately abutting the KO cells vary in size and some cells are elongated perpendicular to the boundary. By contrast, the shapes of WT cells that are distant from the boundary are more regular polygons and the sizes are less varied.

Fig. R3 Co-culture of WT and ZO-1,-2 DKO EpH4 cells stained for ZO-1 (green) and pMLC (red). Scale bar, 20 μ m

Reviewer #1 (Remarks to the Author):

I have no comments.

Reviewer #2 (Remarks to the Author):

Responsive revision; very nice study

Reviewer #3 (Remarks to the Author):

This is the second round of review of this manuscript. The quantification and imaging are well done, and the statistics are robust. The insights gained here should interest the scientific community, placing clearly MAGI scaffolds as apical cortex actin cytoskeleton regulators, ensuring distribution of the tension.

Overall, I am pleased with the answers and precisions given by the authors. There are just a few text points that need to be changed.

1. As suggested in my first review, I feel that some interpretations/conclusions concerning the relations between ZO / MAGI / ASPP are slightly over-interpreted given the data presented (Comment 3-1 in the rebuttal). Even though the authors acknowledge that alternative mechanisms to the linear Afadin/ZO - MAGI - ASPP - PAR3 might exist in their detailed answers (with additional data only shown in the rebuttal that is inconclusive), they still do not discuss it or suggest it clearly in the text. A simple sentence on page 7 stating that MAGIs could also act in parallel to ZOs in the regulation of ROCK activity and/or Par3 recruitment would suffice. Indeed, on page 7 line 2 the authors still write "The parallel between these observations and those in ZO-1,-2 DKO cells suggest that MAGI mediate the antagonism of ROCK that was lost in ZO-1,-2 DKO cells." I understand that the experiments to conclusively answer are long, but at this stage I am just asking that the text and data agree. I have looked at the sentences mentioned in the authors' answers (page 7 line 22 and page 11 line 21), but they fail to convey a clear message.

2. In page 3 in the introduction, authors claim "MAGI contribute to the steady state level of apical domain contractility by modulating junctional localization of aPKC". I fail to see the supporting data. I guess the authors mean ROCK or Par3 instead of aPKC. Indeed they tested directly aPKC only in ASPP mutant conditions and not in MAGI mutants. This sentence should be changed.

3. The physical interaction between ASPP and RASSF8, and that RASSF8 links MAGI and ASPP has been first shown in *Drosophila*. This should be referenced in page 8.

4. Some sentences have been obviously subjected to several rounds of editing, making them hard to understand or illogical and they need to be fixed.

Page 7 line 25: Phosphorylation dynamics at a different residue

Page 9 sentence starting line 33

Page 10 line 25

Finally, I would like to apologise to the authors for the slight delay in my review.

Below are the point-by-point responses to the concerns raised by Reviewer #3.

Comment 1

As suggested in my first review, I feel that some interpretations/conclusions concerning the relations between ZO / MAGI / ASPP are slightly over-interpreted given the data presented (Comment 3-1 in the rebuttal). Even though the authors acknowledge that alternative mechanisms to the linear Afadin/ZO - MAGI - ASPP - PAR3 might exist in their detailed answers (with additional data only shown in the rebuttal that is inconclusive), they still do not discuss it or suggest it clearly in the text. A simple sentence on page 7 stating that MAGIs could also act in parallel to ZOs in the regulation of ROCK activity and/or Par3 recruitment would suffice. Indeed, on page 7 line 2 the authors still write "The parallel between these observations and those in ZO-1,-2 DKO cells suggest that MAGI mediate the antagonism of ROCK that was lost in ZO-1,-2 DKO cells." I understand that the experiments to conclusively answer are long, but at this stage I am just asking that the text and data agree. I have looked at the sentences mentioned in the authors' answers (page 7 line 22 and page 11 line 21), but they fail to convey a clear message.

Response 1

We have made the possibility that "MAGIs could also act in parallel to ZOs" by amending the text as follows:

Page 7 Line 21

It is possible that ZO proteins, afadin and MAGIs act in parallel to modulate Par-3-dependent ROCK activity.

Comment 2

In page 3 in the introduction, authors claim "MAGI contribute to the steady state level of apical domain contractility by modulating junctional localization of aPKC". I fail to see the supporting data. I guess the authors mean ROCK or Par3 instead of aPKC. Indeed they tested directly aPKC only in ASPP mutant conditions and not in MAGI mutants. This sentence should be changed.

Response 2

The sentence in question was changed as follows:

Page 3 Line 23

In this paper, we clarified that the cell adhesion-related molecules MAGI-1 and MAGI-3 contribute to the steady state level of apical domain contractility by mobilizing a complex of signaling proteins

that culminate in aPKC-mediated antagonism of junctional ROCK activity.

Comment 3

The physical interaction between ASPP and RASSF8, and that RASSF8 links MAGI and ASPP has been first shown in *Drosophila*. This should be referenced in page 8.

Response 3

We had cited the study that showed the physical interaction between ASPP and RASSF8 but this specific point was not referenced in our manuscript. We have inserted the following statement, highlighted in red, to the relevant sentence:

Page 8 Line 13

Additionally, ***Drosophila RASSF8 and MAGI interact*** and a loss-of-function RASSF8 mutant that lacks the ability to bind MAGI phenocopies MAGI mutant in *Drosophila* eye development³³.

Comment 4

Some sentences have been obviously subjected to several rounds of editing, making them hard to understand or illogical and they need to be fixed.

Page 7 line 25: Phosphorylation dynamics at a different residue

Page 9 sentence starting line 33

Page 10 line 25

Response 4

The above sentences were edited for clarity as follows:

Page 7 Line 30

Phosphorylation can also regulate Par-3 oligomerization, which is essential for proper localization^{26,27}.

Page 10 Line 3

Therefore, we explored whether ASPP2-PP1 interaction was required in our experimental context. We rescued either wildtype ASPP2 (ASPP2-WTres) or a mutant ASPP2, in which the canonical RVXF motif essential for the binding to PP1 was mutated (KRVF to KARA; ASPP2-KARAs), in ASPP1,2 DKO cells.

Page 10 Line 28

Likewise, application of the 3D projection analysis to ASPP2 rescue cells revealed that aPKC was strongly enriched at GFP-positive junctions in ASPP2-WTres cells but not in ASPP2-KARAs cells (Fig. 4h,i).